# In Vitro Effects of Fentanyl on Aortic Viscoelasticity in a Rat Model of Melatonin Deficiency

**DOI:** 10.3390/ijms25115669

**Published:** 2024-05-23

**Authors:** Andreyan Georgiev, Maria Kaneva, Lyudmila Shikova, Polina Mateeva, Jana Tchekalarova, Mariya Antonova

**Affiliations:** Department of Behavioral Neurobiology, Institute of Neurobiology, Bulgarian Academy of Sciences, 1113 Sofia, Bulgaria; andreyan.ag@gmail.com (A.G.); kanevam@gmail.com (M.K.); lyudmilam@abv.bg (L.S.); mariya.antonova@gmail.com (M.A.)

**Keywords:** viscoelasticity, aorta, pinealectomy, melatonin, fentanyl

## Abstract

Melatonin influences arterial biomechanics, and its absence could cause remodeling of the arterial wall, leading to increased stiffness. Direct effects of fentanyl on the aortic wall have also been observed previously. This study aimed to evaluate in vitro the effects of fentanyl on aortic viscoelasticity in a rat model of melatonin deficiency and to test the hypothesis that melatonin deficiency leads to increased arterial wall stiffness. The viscoelasticity was estimated in strip preparations from pinealectomized (pin, melatonin deficiency) and sham-operated (sham, normal melatonin) adult rats using the forced oscillations method. In the untreated aortic wall pin, the viscoelasticity was not significantly altered. However, combined with 10^−9^ M fentanyl, the pin increased the natural frequency (f_0_) and modulus of elasticity (E’) compared to the sham-operated. Independently, fentanyl treatment decreased f_0_ and E’ compared separately to untreated sham and pin preparations. The effects of fentanyl were neither dose-dependent nor affected by naloxone, suggesting a non-opioid mechanism. Furthermore, an independent effect of naloxone was also detected in the normal rat aortic wall, resulting in reduced E’. Additional studies are needed that may improve the clinical decisions for pain management and anesthesia for certain patients with co-occurring chronic low levels of blood plasma melatonin and some diseases.

## 1. Introduction

Large elastic arteries such as the aorta exhibit nonlinear blood pressure-dependent mechanical wall behavior [1,2,3]. It possesses both elastic and viscous properties at the same time. Such mechanical behavior is referred to as viscoelastic, and its characteristics are often considered as measures of arterial stiffness and compliance. The biomechanical properties of the aorta have been studied extensively over the past few decades using various methods [1,3,4,5]. Without nervous and humoral regulation by the organism, the direct basal viscoelasticity assessment is performed in vitro using partial or cylindrical arterial segments [2,6,7,8,9].

Intrinsic aging of the aortic wall correlates with structural changes in its extracellular matrix [10,11,12]. Reorganization of the extracellular matrix in the form of increased collagen synthesis and elastin degradation, together with smooth muscle cell proliferation, leads to changes in the viscoelasticity of the arterial wall [3,13,14,15]. The latter has been adopted to measure the structural change associated with cardiovascular disease risk [3]. Estimation of viscoelasticity is performed by measuring in vitro two characteristics—the natural frequency (f_0_) and modulus of elasticity (E’) [1]. The natural frequency represents the frequency of the biomaterial at which resonance can occur, with the potential to cause structural damage to the constituent tissue due to the steeply increasing amplitudes of the oscillations. The modulus of elasticity is a measure of arterial stiffness. Viscoelasticity could be used to estimate the effect of opioid analgesics on arterial biomechanics, as these drugs are known to affect arterial blood pressure [16,17].

The pineal hormone melatonin has also been shown to significantly influence the homeostasis of the extracellular matrix of the arterial wall. The neurological and endocrine roles of melatonin have been well described in the literature. However, its effects on arterial viscoelasticity, either direct or indirect, have not been completely studied [18,19,20,21].

Melatonin secretion from the pineal gland is known to decrease with age, both in humans and rodents [22,23]. Apart from this natural process, the clinical presentation of some neurodegenerative, neurological, oncological, metabolic, and rare congenital diseases is inversely correlated with blood plasma melatonin concentration. Decreased melatonin levels are often found in certain types of cancer and some genetic, neurological, or metabolic diseases [24]. Chronic exposure to artificial light at night also suppresses melatonin secretion in the blood plasma [24]. Taking all this into account, pineal dysfunction and the associated melatonin deficiency appear to be not uncommon; therefore, the relationship between aortic wall viscoelasticity and the absence of melatonin hormonal signaling deserves to be investigated in relation to any possible cardiovascular complications following the administration of opioid analgesics before, during or after surgical procedures. One such opioid analgesic is fentanyl, which is commonly used in general anesthesia and the treatment of severe pain.

Rat pinealectomy is a validated model of melatonin deficiency, in which not only plasma melatonin levels are significantly reduced during the day/night cycle but also urinary excretion of its metabolite 6-sulfatoxymelatonin in urine [25,26,27]. In the present work, we aimed to evaluate in vitro the direct effects of fentanyl on viscoelasticity in aortic preparations from sham-operated (sham) rats and in pinealectomized (pin) rats, i.e., rats with induced melatonin deficiency. We also aimed to test the hypothesis that the lack of melatonin leads to increased arterial stiffness in both fentanyl-treated and untreated rat aortic walls.

## 2. Results

### 2.1. Effects of Pinealectomy in Untreated Aortic Preparations

After 30 min initial adaptation (control), the natural frequency (f_0_) in both the sham-operated (sham) and pinealectomized (pin) groups increased linearly with increasing equivalent blood pressure (p) (shown in Figure 1), whereas the dynamic modulus of elasticity (E’) increased exponentially (shown in Figure 2). However, there was no significant difference between the control measurements of the sham and pin groups.

### 2.2. Effects of Pinealectomy in Fentanyl-Treated Aortic Preparations

To further investigate the effects of melatonin deficiency (induced by pinealectomy), f_0_ and E’ were estimated at each fentanyl (Fe) concentration and compared between the sham and pin groups. Natural frequency, as a linear function of equivalent blood pressure, was found to be significantly higher (*p* < 0.035) in the pin group than in the sham group, only at the lowest fentanyl concentration (Fe 10^−9^ M). This difference was estimated to increase with rising equivalent blood pressure. A similar result was obtained for E’ at the same fentanyl concentration (*p* < 0.035). However, higher concentrations of fentanyl did not significantly affect this viscoelastic property. Naloxone (Nal) 10^−6^ M did not change the estimated effects of pinealectomy at higher fentanyl concentrations, as both f_0_ and E’ were significantly higher than in the sham group, but only at 10^−9^ M fentanyl (f_0_, *p* < 0.035; E’, *p* < 0.015). The estimated differences also increased at higher levels of equivalent blood pressure, i.e., >111.8 mmHg (shown in Figure 3 and Figure 4).

### 2.3. Effects of Fentanyl and Fentanyl + Naloxone in the Sham Group

In the sham group, treatment with increasing concentrations of fentanyl resulted in significantly lower values of f_0_ only at the highest 10^−6^ M fentanyl concentration (*p* < 0.035) compared to the control measurement. No other significant differences were found (except for a borderline significance (*p* = 0.054) at 10^−7^ M fentanyl). As with the control measurements, the natural frequency was estimated to increase linearly with increasing equivalent arterial blood pressure (shown in Figure 5).

In contrast to f_0_, E’ was significantly lower (*p* < 0.05) at all fentanyl concentrations compared to the sham group control. However, measurements taken after the 30-min wash-out, following the last concentration of fentanyl applied, resulted in an almost complete recovery of the modulus of elasticity to control levels. Furthermore, E’ again was significantly reduced after the subsequent 10-min treatment with 10^−6^ M naloxone solution (*p* < 0.015). Subsequent treatment with all incremental concentrations of fentanyl, in combination with 10^−6^ M naloxone, kept the modulus of elasticity significantly lower (*p* < 0.015) than the control measurement in the sham group, except at 10^−8^ M fentanyl (shown in Figure 6).

### 2.4. Effects of Fentanyl and Fentanyl + Naloxone in the Pin Group

In the pin group, the comparison between the different treatments and the control measurement shows that f_0_ was significantly lower only after the 30-min wash-out (*p* < 0.015), following the last applied concentration of fentanyl, then after the subsequent 10 min of treatment with 10^−6^ M naloxone (*p* < 0.015), and again after 10^−6^ M fentanyl + 10^−6^ M naloxone (*p* < 0.035) (shown in Figure 7). E’ was also estimated to be significantly lower when compared to the control measurement in the pin group at 10^−7^ M fentanyl (*p* < 0.035), after the 30-min wash-out (*p* < 0.035), after the last applied concentration of fentanyl, then after the subsequent 10 min treatment with 10^−6^ M naloxone (*p* < 0.015) (shown in Figure 8).

## 3. Discussion

### 3.1. Effects of Pinealectomy

In untreated aortic preparations from pinealectomized and sham-operated rats, the observed differences in f_0_ and E’ were not statistically significant. These results suggest that melatonin deficiency alone may not significantly alter the aortic wall viscoelasticity. The lack of significance is probably due to the small sample size, and the effect of pinealectomy needs to be confirmed in a large sample study. However, when comparing the two groups in the presence of fentanyl, the values of f_0_ and E’ remain higher in the pin group even after their reduction by the fentanyl treatment.

The increased aortic wall stiffness in the pin group (despite the presence of a low concentration of fentanyl 10^−9^ M) is consistent with the expected fibrotic rebuilding due to induced melatonin deficiency, associated with increased content of collagen types I and III in the media, as reported by Repová-Bednárová et al. in a continuous light exposure rat model of melatonin deficiency [20]. Repová-Bednárová et al. have demonstrated that melatonin treatment reduced the effects of pineal gland dysfunction induced by 24-h light exposure [20]. Here, we suggest that our results are complementary as they indicate that the pinealectomy model of melatonin deficiency may also lead to aortic wall remodeling and increased stiffness (i.e., decreased distensibility).

### 3.2. Effects of Fentanyl

In the sham group, fentanyl treatment resulted in significantly lower values for both the natural frequency and the modulus of elasticity compared to the control measurement in the same group. For f_0_, this was true only at 10^−6^ M, whereas for E’, the effect was significant over the whole range of fentanyl concentration. However, a tendency to decrease f_0_ was observed at other concentrations with a borderline significance (*p* < 0.054–0.075). A further increase in the sample tested will show whether the difference is significant. The addition of naloxone in the sham group did not reverse the effect of fentanyl, except at 10^−8^ M fentanyl. Similar but less clear results were also obtained within the pin group, where f_0_ values were significantly lower than the pin control only at 10^−6^ M fentanyl in the presence of 10^−6^ M naloxone, and E’ values were significantly lower than the pin control only at 10^−7^ M fentanyl.

It could be concluded that due to fentanyl treatment, lower values of f_0_ and E’ indicate a risk of reaching noxious oscillations at physiologically normal heart rate or lower external frequencies and a reduced arterial wall stiffness (i.e., increased distensibility) compared to untreated preparations. The slight relaxant effect of fentanyl in sham preparations is consistent with similar conclusions presented in the literature [17]. The pharmacological effects of fentanyl include arterial hypotension [17]. There is evidence for opioid receptor expression in arterial wall cells, such as endothelial and vascular smooth muscle cells, where the effects of opioid agonists on contractility have been reported [28,29,30,31,32,33]. In addition, other in vitro studies have shown that fentanyl (10^−6^ M) exerts a direct relaxant effect on phenylephrine-precontracted rat aorta, which is endothelium-independent and due to antagonism of α_1D_-adrenergic receptors on vascular smooth muscle cells [34,35]. Karasawa et al. have found that fentanyl-induced relaxation is mediated by alpha-adrenergic receptors and modulated only by the endothelium [34]. The same authors reported similar results for sufentanil and alfentanil, which act directly on the smooth muscle [36]. In our study, the endothelium was removed and did not affect smooth muscle viscoelasticity. We also found a similar relaxing effect of fentanyl in pin preparations. However, the resulting relaxation cannot restore the viscoelastic properties of the pin aorta wall to those of the sham aorta wall. Subsequent treatment with naloxone did not significantly reverse the effects of fentanyl, suggesting a possible non-opioid mechanism of action of fentanyl.

After almost complete recovery of the modulus of elasticity in the sham group after a 30-min wash-out, naloxone treatment led to a significant decrease of E’. This effect was not expected given its known opioid antagonism, which would suggest an opposed effect to that seen with fentanyl, i.e., an increase of E’. Recently, Migheli et al. have shown that naloxone reduces the excessive intercellular production of reactive oxygen species (ROS) in rat pheochromocytoma PC12 cells [37]. This finding provides a possible link between the known interlayer ROS signaling in the vascular wall and the effect of naloxone observed in our study [38]. Further investigation of the independent effect of naloxone without opioid pretreatment would be beneficial to clarify these findings. In contrast, no independent effect of naloxone was observed in the pin group, while the fentanyl-induced decrease in E’ was not reversed by either the 30-min wash-out or the subsequent treatment with naloxone. A possible explanation for this observation is that fentanyl may not be thoroughly washed out in pin preparations. This suggestion is consistent with our empirical observation that aortic preparations from pinealectomized rats show an increased fat accumulation at the macroscopic level compared to normal (sham) preparations. In addition, according to Sutcliffe et al., the higher lipophilicity of fentanyl leads to increased deposition in the cellular lipid membrane. Such a theory may explain the reduced ability of fentanyl to be washed out and its effect to be antagonized by naloxone [39].

## 4. Materials and Methods

### 4.1. Materials

The use of opioid analgesics was permitted with License No. 5/19.04.2021 of the Bulgarian Ministry of Health. Fentanyl was supplied as fentanyl citrate 50 µg/mL solution from UAB Santonika, Lithuania, and naloxone was supplied as naloxone hydrochloride 0.4 mg/mL solution from Warsaw Pharmaceutical Works Polfa, Poland. The initial fentanyl solution was diluted with a nutrient medium and used in 4 solutions to increase concentration—10^−9^, 10^−8^, 10^−7^, and 10^−6^ M (mol/L). The initial naloxone solution was diluted to 10^−6^ M.

### 4.2. Animals

The rat aorta is widely used in physiological and pharmacological studies. The rat aorta is a good analog of the human aorta because of the similarity between the biomechanical properties and the physiological responses. The main advantage of using rat aorta is that it can be prepared and studied immediately after death with preserved vitality, which is difficult with human tissues. All this allows the study of viscoelasticity in a living tissue (as the applied method) without the influence of the regulating mechanisms of the organism.

All experiments were conducted with 4-month-old male Wistar rats grown on wood chip bedding. The rats were supplied in-house from the Experimental and Breeding Base for Experimental Animals, part of the Institute of Neurobiology, Bulgarian Academy of Sciences.

### 4.3. Surgical Procedure

The rats were divided into two groups according to the surgical manipulation: sham-operated (sham, n = 5) and rats with pinealectomy (pin, n = 5). Pinealectomy was performed according to the protocol of Hoffmann and Reiter and our previous studies [40,41,42]. The pineal gland was removed with fine forceps in anesthetized rats (ketamine 40 mg/kg, i.p. and xylazine 20 mg/kg, s.c.) fixed to the stereotaxic apparatus (Stoelting, Europe, Dublin, Ireland). After the surgery, routine manipulations were applied, and rats were injected with Ringer’s solution and gentamicin as an antibiotic for three days. The sham group underwent similar procedures, except for the removal of the gland.

Recently, we reported that plasma melatonin levels in sham-operated rats have a circadian pattern with a peak during the dark period and a nadir during the light period [42]. In addition, we also reported that rats with pinealectomy had a flattened 24-h pattern of plasma melatonin [42]. These findings agree with Lewy et al., who reported a nocturnal plasma melatonin level of 52 pg/mol in sham-operated animals. In contrast, in pinealectomized, the hormone was below the minimum detectable concentration (<1 pg/mol) 7 days after surgery [27]. The authors have also confirmed that the pineal gland is the primary source of the hormone. At the same time, several other tissues, including the gastrointestinal tract, release the hormone with mainly paracrine functions [27].

Recently, we reported that the absence of the pineal gland in young adult rats of the same age had a detrimental effect on some physiological, metabolic, and biochemical parameters [43]. Melatonin deficiency was also found to accelerate the aging process in young adult animals, increasing arterial blood pressure, blood glucose, and triglyceride levels [43,44,45,46].

In another model of melatonin deficiency, a continuous light exposure model, 3-month-old Wistar rats demonstrated the development of hypertension and adaptive remodeling of the thoracic aorta associated with significant accumulation of collagen I and III in the medial layer. Compared to untreated animals, melatonin administration reduced blood pressure and collagen accumulation [20]. However, in addition to the lack of sympatholytic effect of melatonin, arterial remodeling in the continuous light-exposure model could also be attributed to increased activity of the hypothalamic-pituitary-adrenal axis [47].

### 4.4. Preparation of Aortic Strips

All experimental animals were sacrificed by guillotine after CO_2_ inhalation 14 days after surgery. A portion of the aorta was dissected between the end of the aortic arch and an area slightly above the aortic hiatus of the diaphragm, including most of the descending thoracic aorta. The aortic segment was then immersed in a cold (4 °C) nutrient medium and cleaned under a microscope to remove both the adventitial and endothelial layers. The latter was removed by gently rubbing the inner surface of the artery with a thin stainless-steel wire in a longitudinal and circular motion for a few seconds. The remaining tunica media layer was spirally cut into a helical strip preparation approximately 3 mm wide. The procedure was performed in such a way as to avoid any stretching of the strip and to ensure that the viscoelastic properties were not compromised.

In recent years, new studies have expanded the knowledge of PVAT and its role in regulating arterial smooth muscle tone and extracellular matrix composition [47,48].

Melatonin MT_1_ receptors have been found in rat mesenteric artery’s adventitia and media layers [49,50]. Only the MT_1_ type has been confirmed in rat aorta, where these receptors have been detected at mRNA and protein levels predominantly in the outermost layer—the tunica adventitia [50]. The authors suggest that melatonin receptors do not directly contribute to aortic contractility. In our study, the adventitia is removed during the preparation of aortic strips, allowing the direct effects of fentanyl on the intrinsic viscoelasticity of the medial layer to be assessed.

### 4.5. Nutrient Medium

A nutrient medium was used throughout all experiment durations so that the vitality of each strip preparation would remain preserved while being kept in quasi-physiological conditions. The nutrient medium was a modified Tyrode solution with pH 7.2–7.4, consisting of the following substances in millimolar concentrations: 136.9 NaCI, 2.7 KCI, 2.0 CaCI_2_, 0.6 MgCI_2_, 11.9 NaHCO_3_, 0.5 NaH_2_PO_4_, and 11.5 glucose, continuously aerated with a carbogen gas mixture (95% CO_2_, 5% O_2_) at 37 °C.

The experimental protocol is further described in Section 4.7. below.

### 4.6. Forced Oscillations Method

The forced oscillations method applies the Kelvin-Voigt viscoelasticity model, where the preparation is represented by both a viscous damper and an elastic spring connected in parallel (shown in Figure 9). Low-frequency sinusoidal oscillations with constant amplitude (excitation oscillations) are applied at the upper end of the suspended strip preparation. The response oscillations of the lower end of the strip are known as forced oscillations. When the frequency of excitation oscillations sweeps a chosen interval up and down, so does the frequency of forced oscillations, but their amplitude rises and diminishes, going through the resonance. The forced oscillations are recorded simultaneously with applied excitation oscillations by a computer’s analog-to-discrete converter (ADC). The method is described in detail in Antonova et al., 2024 [51].

The graphical representation of the dependence of the amplitude of the forced oscillations on the frequency is known as the resonance curve (shown in Figure 10). In contrast to the linearly elastic materials, where the resonance curve is symmetrical, the arterial wall exhibits nonlinear elasticity represented as bending the “bone” of the resonance curve to the left (softening-type nonlinear elasticity) or to the right (hardening-type nonlinear elasticity) [52].

Four lead rings with known mass are consecutively placed over the mask, creating elongations of the preparation like that made by the intraluminal pressure in the artery. This way, the strip preparation stress state could be assumed to be very close to the stress state in a cylindrical arterial segment under intraluminal pressure (named here equivalent blood pressure, mmHg). The obtained resonance curves (shown in Figure 10) are used to measure the 3 dB bandwidth, the octave length, and the natural frequency (f_0_), as explained in previous studies [53].

The dynamic modulus of elasticity (E’) is calculated using the following equation, considering the preparation’s geometry [51,53].
E’ = (2 × π × f_0_)^2^ × m × (L_0_ + ∆L)/S(1)
where:

ΔL—elongation of the preparation

L_0_—initial length of the preparation

S—cross-section area of the preparation

### 4.7. Experimental Protocol

Once suspended, the preparation was allowed to adapt to the experimental conditions for 30 min while superfusing with nutrient medium (6 mL/min). After the adaptation period had elapsed, control measurements were taken. The superfusion was stopped during the record of oscillations to avoid the attachment of an additional oscillating mass by the solution. Each subsequent measurement was taken with 1 out of 4 separate suspended concentrated masses, starting with the lightest concentrated mass in an incremental order. Each mass creates a stretch corresponding to a discrete level of the equivalent intraluminal pressure between 77.2 and 157.4 mmHg. The same procedure was repeated with each of the five concentrations of fentanyl in incremental order, starting with 10^−9^ M added to the nutrient medium (named further “fentanyl solution”)—filled in a second thermostatic glass. Each higher-concentration solution was then filled in the same thermostatic glass after emptying it from the previous solution. No washing of the preparation was performed after each concentration application so that each subsequent concentration of fentanyl is additive for a cumulative effect on the preparation. After all fentanyl solutions had been used, the preparation was washed out, and it was superfused with a pure nutrient medium for another 30 min. After this period, experiments were performed again at four equivalent pressures to control the elimination of the fentanyl. Then, the preparation was superfused for 10 min with naloxone 10^−6^ M added to the nutrient medium. Another four records were made for control of the naloxone effect. It is sought naloxone to bind the opioid receptors and to preserve them against bounding the fentanyl in the next step. The experimental procedure was repeated with each of the previous fentanyl solutions but in addition to naloxone (shown in Figure 11).

### 4.8. Data Analysis

The recorded signals of excitation and forced oscillations were processed digitally with OriginPro version 9.0 (OriginLab, Northampton, MA, USA) and Excel 2019 version 2404 (Microsoft Corp., Redmond, WA, USA). GraphPad Prism version 6.01 (GraphPad Software, Inc., La Jolla, CA, USA) was used for statistical analysis. Linear and nonlinear regression were used to estimate f_0_ and E’, respectively. Slopes and elevations/intercepts (for f_0_) and regression curves (for E’) were compared for differences, and statistical significance was assumed if *p* < 0.05.

## 5. Conclusions

In conclusion, two viscoelastic properties of the tunica media of the aortic wall have been quantitatively estimated, in contrast to other studies, where only a qualitative estimate of the elastic modulus is made. Our results suggest that in the untreated rat aortic wall, melatonin deficiency alone may not significantly alter the natural frequency and stiffness of the isolated medial layer. Fentanyl directly impacts aortic wall viscoelasticity by decreasing the arterial stiffness and natural frequency in both normal rats and rats with a chronic melatonin deficiency. This effect shifts the risk of noxious oscillations at frequencies close to the decreased natural frequency. Nevertheless, melatonin deficiency in combination with 10^−9^ M fentanyl leads to higher natural frequency and arterial stiffness than the combination of the same fentanyl treatment and normal rat aortic wall. The observed effects of fentanyl were neither concentration-dependent nor affected by the opioid antagonist naloxone, indicating that these effects are achieved mainly due to activation of receptors other than the opioid. In addition, an unexpected independent effect of naloxone was also detected on the normal rat aortic wall, leading to reduced arterial stiffness. This research provides new evidence that fentanyl acts by a non-opioid mechanism as a direct vasorelaxant at the arterial wall medial layer, in support of the already known hypotensive action in vivo. Further investigation is needed to clarify the underlying mechanisms, which may improve clinical decisions in pain management and anesthesia with fentanyl for certain patients with co-occurring chronic low levels of blood plasma melatonin and some cancer, genetic, neurological, or metabolic diseases.

## Figures and Tables

**Figure 1 ijms-25-05669-f001:**
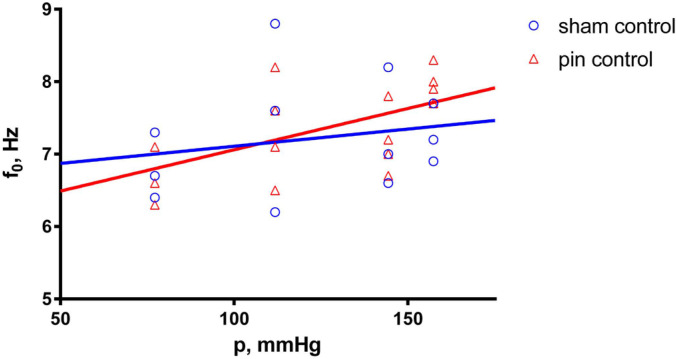
Natural frequency (f_0_) as a function of the equivalent blood pressure (p). Effect comparison between the sham (n = 3) and pin (n = 4) groups, with measurements taken after the 30 min adaptation period (control). Solid lines show the best-fit regression model (blue line—sham group; red line—pin group).

**Figure 2 ijms-25-05669-f002:**
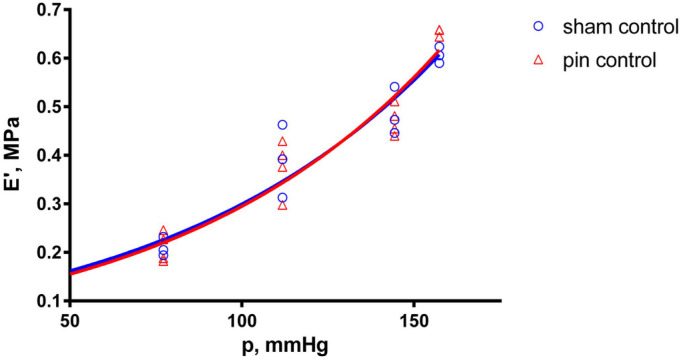
Modulus of elasticity (E’) as a function of the equivalent blood pressure (p). Effect comparison between the sham (n = 3) and pin (n = 4) groups, with measurements taken after the 30 min adaptation period (control). Solid curves show the best-fit regression model (blue curve—sham group; red curve—pin group).

**Figure 3 ijms-25-05669-f003:**
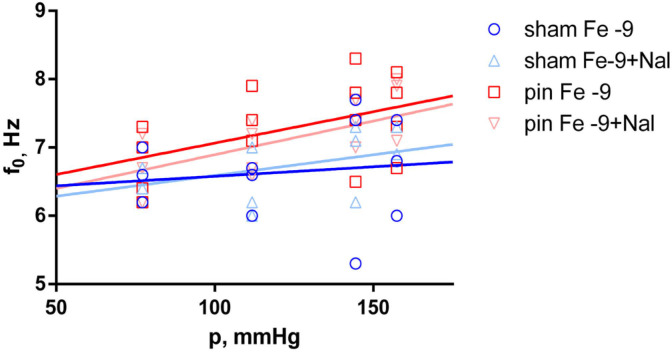
Natural frequency (f_0_) as a function of the equivalent blood pressure (p). Significant effect comparison between the sham (n = 3) and pin (n = 4) groups in equal concentrations of fentanyl—10^−9^ M fentanyl (Fe −9) and 10^−9^ M fentanyl + 10^−6^ M naloxone (Fe −9 + Nal) treatment. Solid lines show the best-fit regression model (dark blue line—sham group with Fe −9; light blue line—sham group with Fe −9 + naloxone; dark red line—pin group with Fe −9; light red line—pin group with Fe −9 + naloxone).

**Figure 4 ijms-25-05669-f004:**
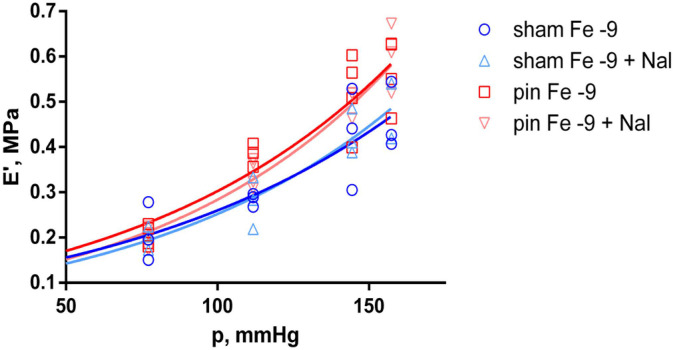
Modulus of elasticity (E’) as a function of the equivalent blood pressure (p). Significant effect comparison between the sham (n = 3) and pin (n = 4) groups in equal concentrations of fentanyl—10^−9^ M fentanyl (Fe −9) and 10^−9^ M fentanyl + 10^−6^ M naloxone (Fe −9 + Nal) treatment. Solid curves show the best-fit regression model (dark blue curve—sham group with Fe −9; light blue curve—sham group with Fe −9 + naloxone; dark red curve—pin group with Fe −9; light red curve—pin group with Fe −9 + naloxone).

**Figure 5 ijms-25-05669-f005:**
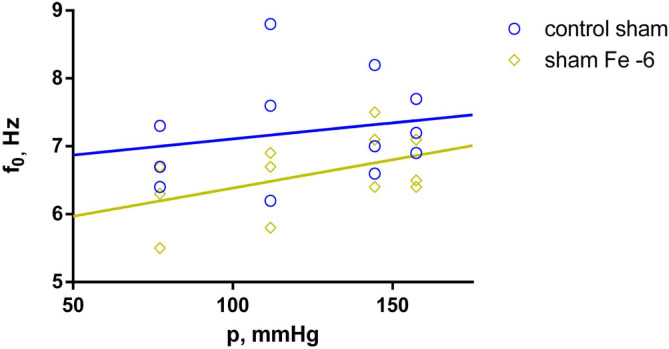
Natural frequency (f_0_) as a function of the equivalent blood pressure (p). Significant effect comparison within the sham group (n = 3), with measurements taken after control and fentanyl 10^−6^ treatment (Fe −6). Solid lines show the best-fit regression model (blue line—sham group control; dark yellow line—sham group with Fe −6).

**Figure 6 ijms-25-05669-f006:**
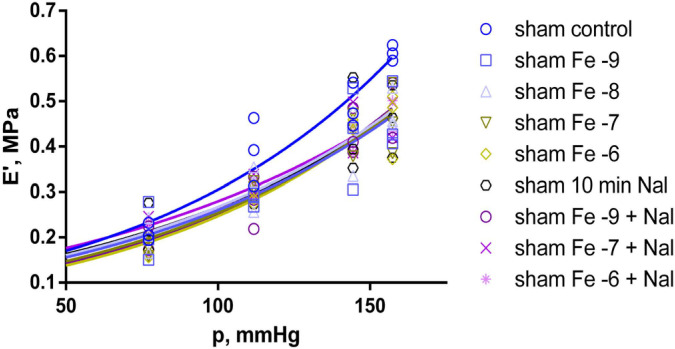
Modulus of elasticity (E’) as a function of the equivalent blood pressure (p). Significant effect comparison within the sham group (n = 3), with measurements taken after control, fentanyl 10^−9^, 10^−8^, 10^−7^ and 10^−6^ M (Fe −9, Fe −8, Fe −7 and Fe −6), 10 min 10^−6^ M naloxone (10 min Nal), and fentanyl 10^−9^, 10^−7^ and 10^−6^ M + 10^−6^ M naloxone(Fe −9 + Nal, Fe −7 + Nal, Fe −6 + Nal) treatment. Solid curves show the best-fit regression model (dark blue curve—sham group control; blue curve—sham group with Fe −9; light blue curve—sham group with Fe −8; dark yellow curve—sham group with Fe −7; light yellow curve—sham group with Fe −6; black curve—sham group after 10 min 10^−6^ M naloxone; dark magenta curve—sham group with Fe −9 + naloxone; magenta curve—sham group with Fe −7 + naloxone; light magenta curve—sham group with Fe −6 + naloxone).

**Figure 7 ijms-25-05669-f007:**
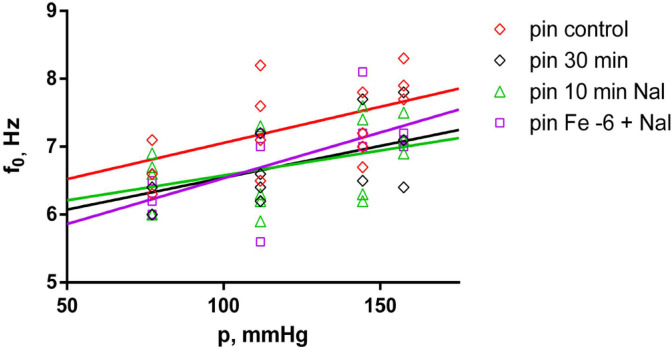
Natural frequency (f_0_) as a function of the equivalent blood pressure (p). Significant effect comparison within the pin group (n = 4), with measurements taken after control, 30 min wash-out, 10 min 10^−6^ M naloxone (10 min Nal) and 10^−6^ M fentanyl + 10^−6^ M naloxone treatment (Fe −6 + Nal). Solid lines show the best-fit regression model (red line—pin group control; black line—pin group after 30 min wash-out; green line—pin group after 10 min naloxone; magenta line—pin group with Fe −6 + naloxone).

**Figure 8 ijms-25-05669-f008:**
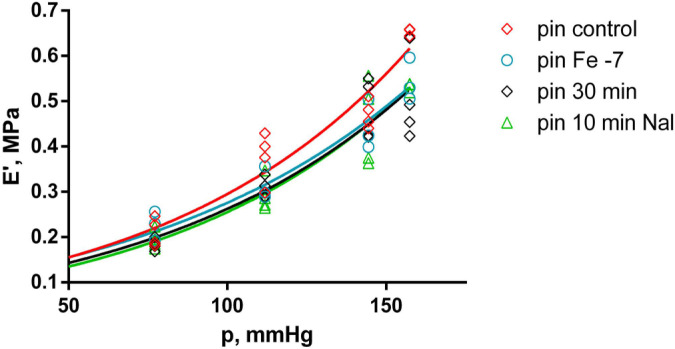
Modulus of elasticity (E’) as a function of the equivalent blood pressure (p). Significant effect comparison within the pin group (n = 4), with measurements taken after control, 10^−7^ M fentanyl (Fe −7), 30 min wash-out, and 10 min 10^−6^ M naloxone (10 min Nal) treatment. Solid curves show the best-fit regression model (red curve—pin group control; light blue curve—pin group with Fe −7; black curve—pin group after 30 min wash-out; green curve—pin group after 10 min naloxone).

**Figure 9 ijms-25-05669-f009:**
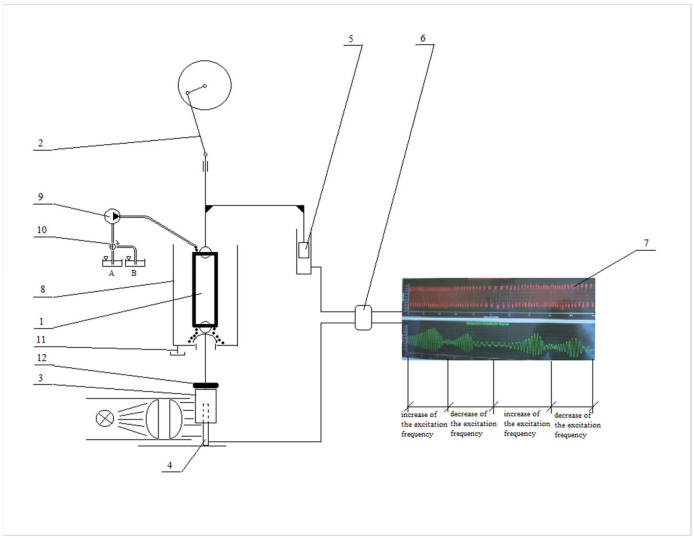
Scheme of the forced oscillations method: 1. Strip preparation, 2. Mechanism for generation of sinusoidal oscillations, 3. Opaque mask, 4. Photo element, 5. Electronic transducer, 6. Analog-to-discrete converter (ADC), 7. Signals display, 8. Organ chamber, 9. Peristaltic pump, 10. Valve, 11. Orifice, 12. Lead ring (concentrated mass), A—solution container no. 1, B—solution container no. 2.

**Figure 10 ijms-25-05669-f010:**
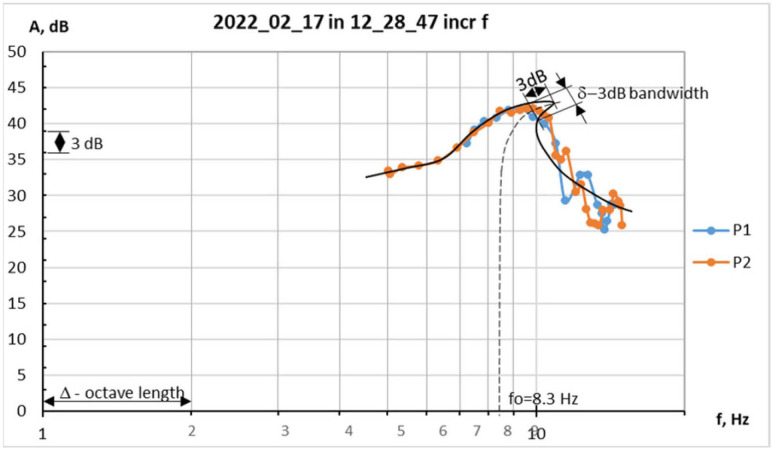
Theoretical resonance curve (solid black) based on two experimental resonance curves (P1 and P2). f_0_—natural frequency.

**Figure 11 ijms-25-05669-f011:**
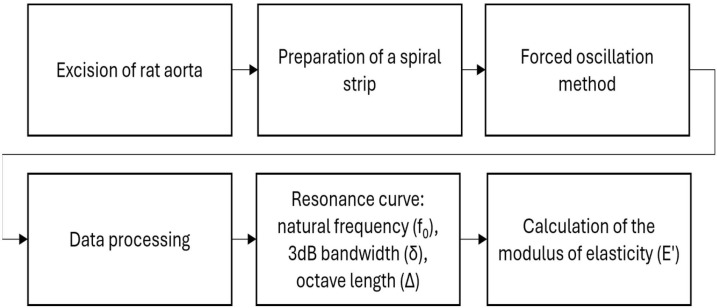
Experimental protocol.

## Data Availability

All data generated or analyzed during this study are included in this article. Further inquiries can be directed to the corresponding author.

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
