# Peer review of "In Vitro Effects of Fentanyl on Aortic Viscoelasticity in a Rat Model of Melatonin Deficiency"

_ijms, 2024, doi:10.3390/ijms25115669_

Round 1
Reviewer 1 Report
Comments and Suggestions for Authors
In this manuscript, Georgiev et. al investigate the effects of fentanyl on the aortic viscoelasticity of adult rats having been pinealectomized to serve as a model system for melatonin deficiency. Two major experimental parameters were used to assess aortic viscoelasticity: the natural frequency (f0) and the dynamic modulus of elasticity (E ́). The Authors found that melatonin deficiency alone does not significantly alter the aortic wall viscoelasticity. Fentanyl treatment in rat models without melatonin deficiency resulted in lower f0 and E ́. However, with fentanyl treatment, the melatonin deficient group showed significantly increased f0 and E ́. Additionally, treatment of naloxone (an opioid treatment medication) did not recover the viscoelasticity change caused by fentanyl. Ultimately, the Authors saw that in the melatonin deficient rat group, the untreated aortic wall did not significantly alter viscoelasticity.
While the data appears solid and the results are well summarized, the background information and significance of the present study are lacking. More efforts need to be made to present the scientific impact of the work or to at least provide needed context to justify the experimental approach and the model system. These will be necessary to present the conclusions of the study in a manner that readers can appreciate. Therefore, I recommend the manuscript be accepted only after the following specific concerns are addressed:
Specific comments to be addressed:
1. In general, the Authors need to elaborate more on the significance of studying the effects of fentanyl and naloxone with/without Melatonin deficiency. Also, the Authors need to present more background earlier in the manuscript regarding the use of f0 and E ́ in assaying heart viscoelasticity. The Authors’ goal and hypothesis were not clearly stated.
2. The Authors need to address the discrepancies of modelling viscoelasticity using rat heart as compared to humans. Yes, it is difficult to model these effects otherwise, but some discussion or brief mention of justifying the animal model is needed.
3. Is there literature to support the claim in Line 196?
4. The discussion section does not directly reference the results presented in section 2, which makes the manuscript hard to follow.
5. Figure legends are not very informative.
6. In lines 200-202, the Authors imply that the reason the pin group has higher values of f0 and E ́ than the sham with treatment of fentanyl is that the effect of pinealectomy is dominant. Is there any literature precedent or rationale to support or explain your claim? Please refrain from speculation with no reference to previous works or without a solid hypothesis.
7. The present manuscript lacks a clear conclusion. Currently, the end of the discussion section reads as a basic summary of experimental observations with no context: "This study provides further evidence of the effects of melatonin deficiency on aortic viscoelasticity in the presence of an opioid analgesic such as fentanyl, which is commonly used in general anesthesia and the treatment of severe pain". It is not immediately clear what are the major advances or novelty in the work. The Authors must clearly discuss the scientific implication of their results and provide context to the usefulness of the research presented.
Author Response
Specific comments to be addressed:
Point #1. In general, the Authors need to elaborate more on the significance of studying the effects of fentanyl and naloxone with/without Melatonin deficiency. Also, the Authors need to present more background earlier in the manuscript regarding the use of f0 and E ́ in assaying heart viscoelasticity. The Authors’ goal and hypothesis were not clearly stated.
Response: The importance of studying the effects of fentanyl and naloxone with or without melatonin deficiency is expanded at the end of the Introduction section in the revised manuscript.
In this study, we evaluated the viscoelasticity of the medial layer of the thoracic aorta, which is composed of a different type of muscle (smooth) than the heart (cardiac). There is also a difference in the composition of the extracellular matrix, which plays a significant role in the overall viscoelasticity. However, more background on f0 and E ́ is provided in the manuscript, i.e. in the Introduction section.
Point #2. The Authors need to address the discrepancies of modelling viscoelasticity using rat heart as compared to humans. Yes, it is difficult to model these effects otherwise, but some discussion or brief mention of justifying the animal model is needed.
Response: We have previous publications on the viscoelasticity of rat aorta in vitro [1, 3, 54]. The rat aorta is widely used in physiological and pharmacological studies. The rat aorta is a good analog of the human aorta because of the similarity between the biomechanical properties and the physiological responses. The main advantage of using rat aorta is that it can be prepared and studied immediately after death with preserved vitality, which is difficult with human tissues. All this allows the study of viscoelasticity in a living tissue (as the applied method) without the influence of the regulating mechanisms of the organism. As mentioned in the Introduction section, rat pinealectomy is a validated model of melatonin deficiency.
A new text is added in the subsection 4.2. Animals.
Point #3. Is there literature to support the claim in Line 196?
Response: No, we are not aware of any previous evidence in the literature for the observed differences in f0 and E' of untreated aortic tissue obtained from sham-operated (sham) and pinealectomized (pin) rats. Although not statistically significant, we reported our observational evidence for changes in the and E, probably due to the small sample size. This claim is now withdrawn from the revised manuscript (lines 139 and 140) as it will be the subject of another manuscript presenting results for a larger sample and only from the untreated aortic preparations.
Point #4. The discussion section does not directly reference the results presented in section 2, which makes the manuscript hard to follow.
Response: In fact, headings 2.1 and 2.2 were discussed in 3.2 - Effects of Pinealectomy, and headings 2.3 and 2.4 were discussed in 3.1 - Effects of Fentanyl. In the revised manuscript, headings 3.1 and 3.2 are now swapped, along with their corresponding text paragraphs, so that the Discussion section refers directly to the results as follows:
2.1 and 2.2 are discussed in 3.1 - Effects of pinealectomy
2.3 and 2.4 are discussed in 3.2 - Effects of Fentanyl
Please note that for clarity only the headings in section 3 are highlighted and not the whole paragraphs.
Point #5. Figure legends are not very informative.
Response: Figure legends now include explanations of abbreviations. In addition, some errors in the relationship between legends and corresponding figures have been corrected in Figures 7 and 8.
Point #6. In lines 200-202, the Authors imply that the reason the pin group has higher values of f0 and E ́ than the sham with treatment of fentanyl is that the effect of pinealectomy is dominant. Is there any literature precedent or rationale to support or explain your claim? Please refrain from speculation with no reference to previous works or without a solid hypothesis.
Response: There is no literature precedent or rationale to support our claim, so this sentence has been removed from the revised manuscript.
Point #7. The present manuscript lacks a clear conclusion. Currently, the end of the discussion section reads as a basic summary of experimental observations with no context: "This study provides further evidence of the effects of melatonin deficiency on aortic viscoelasticity in the presence of an opioid analgesic such as fentanyl, which is commonly used in general anesthesia and the treatment of severe pain". It is not immediately clear what are the major advances or novelty in the work. The Authors must clearly discuss the scientific implication of their results and provide context to the usefulness of the research presented.
Response: The scientific implications of our findings and support for research utility are now included in the revised manuscript's expanded Conclusions section. The novelty of the work is also clarified in the same section.

Reviewer 2 Report
Comments and Suggestions for Authors
This study investigates the impact of fentanyl on aortic viscoelasticity in a rat model with melatonin deficiency. Although the study offers valuable insights, it also exhibits some weaknesses, for which corresponding improvement suggestions are provided:
1. Hypotheses and Objectives Clarification: The study aims to assess the impact of fentanyl on aortic viscoelasticity in rats with melatonin deficiency, with a secondary aim of investigating whether melatonin deficiency contributes to arterial stiffness. There is a disconnect between fentanyl and melatonin deficiency. In clinical scenarios, under what circumstances do fentanyl and melatonin deficiency coexist? Why is this research significant and necessary? The objectives lack clarity, and the hypothesis is somewhat ambiguous. To improve clarity, explicitly outlining the primary and secondary objectives, along with a concise, testable hypothesis, would be advantageous.
2. Please include trichrome staining and Verhoeff Van Gieson staining results from the continuous light exposure model of melatonin deficiency in 3-month-old Wistar rats to demonstrate the model's development of hypertension and its impact on the thoracic aorta, including significant accumulation of collagen I and III in the medial layer and breakdown of elastin fibers. Compared to untreated animals, administration of melatonin reduced blood pressure and collagen accumulation.
3. Please clarify the terms "Pin" and "pinealectomy." I cannot find "Pin." Please provide the full name or explanation when abbreviations are first introduced throughout the manuscript. For instance, Fe? P? f0 and E´….? Also, in the figures.
Comments on the Quality of English Language3. Please clarify the terms "Pin" and "pinealectomy." I cannot find "Pin." Please provide the full name or explanation when abbreviations are first introduced throughout the manuscript. For instance, Fe? P? f0 and E´….? Also, in the figures.
Author Response
Reviewer 2:
This study investigates the impact of fentanyl on aortic viscoelasticity in a rat model with melatonin deficiency. Although the study offers valuable insights, it also exhibits some weaknesses, for which corresponding improvement suggestions are provided:
Point #1. Hypotheses and Objectives Clarification: The study aims to assess the impact of fentanyl on aortic viscoelasticity in rats with melatonin deficiency, with a secondary aim of investigating whether melatonin deficiency contributes to arterial stiffness. There is a disconnect between fentanyl and melatonin deficiency. In clinical scenarios, under what circumstances do fentanyl and melatonin deficiency coexist? Why is this research significant and necessary? The objectives lack clarity, and the hypothesis is somewhat ambiguous. To improve clarity, explicitly outlining the primary and secondary objectives, along with a concise, testable hypothesis, would be advantageous.
Response: The revised manuscript clarifies the hypotheses and aims in the Introduction section. In the Discussion section, some of the text explaining the clinical scenarios in which fentanyl and melatonin deficiency coexist has been expanded and moved to the end of the Introduction section.
Point #2. Please include trichrome staining and Verhoeff Van Gieson staining results from the continuous light exposure model of melatonin deficiency in 3-month-old Wistar rats to demonstrate the model's development of hypertension and its impact on the thoracic aorta, including significant accumulation of collagen I and III in the medial layer and breakdown of elastin fibers. Compared to untreated animals, administration of melatonin reduced blood pressure and collagen accumulation.
Response: We used another model of melatonin deficiency with removal of the pineal gland rather than constant light exposure. We did not study the effect of melatonin deficiency induced by pinealectomy on blood pressure and morphological changes in histological preparations. The aortic preparation was completely used for the forced oscillation method. Therefore, we have removed the paragraph in the Materials and Methods section (page 8, lines 265-271), which is not closely related to the methods and the question considered in our study.
Point #3. Please clarify the terms "Pin" and "pinealectomy." I cannot find "Pin." Please provide the full name or explanation when abbreviations are first introduced throughout the manuscript. For instance, Fe? P? f0 and E´….? Also, in the figures.
Response: The terms "pin" and "pinealectomy" are defined in Section 4, Materials and Methods:
"...4.3. Surgical procedure
The rats were divided into two groups according to the surgical manipulation: sham-operated rats (sham, n=5) and pinealectomized rats (pin, n=5). ..."
The revised manuscript now includes further explanations and abbreviations, both in the text and in the figure legends.
